# Linear-regression-based algorithms can succeed at identifying microbial functional groups despite the nonlinearity of ecological function

**Yuanchen Zhao** [1], **Otto X. Cordero** [2], **Mikhail Tikhonov** [3] *

**1** School of Physics, Nanjing University, Nanjing, Jiangsu, the People's Republic of China, **2** Department of Civil and Environmental Engineering, Massachusetts Institute of Technology, Cambridge, Massachusetts, United States of America, **3** Department of Physics, Washington University in St. Louis, St. Louis, Missouri, United States of America

* tikhonov@wustl.edu

**Data Availability Statement:** All relevant data are within the manuscript and its Supporting information files. The MATLAB code (Mathworks,

## Abstract

Microbial communities play key roles across diverse environments. Predicting their function and dynamics is a key goal of microbial ecology, but detailed microscopic descriptions of these systems can be prohibitively complex. One approach to deal with this complexity is to resort to coarser representations. Several approaches have sought to identify useful group-ings of microbial species in a data-driven way. Of these, recent work has claimed some empirical success at *de novo* discovery of coarse representations predictive of a given func-tion using methods as simple as a linear regression, against multiple groups of species or even a single such group (the ensemble quotient optimization (EQO) approach). Modeling community function as a linear combination of individual species' contributions appears sim-plistic. However, the task of identifying a predictive coarsening of an ecosystem is distinct from the task of predicting the function well, and it is conceivable that the former could be accomplished by a simpler methodology than the latter. Here, we use the resource competi-tion framework to design a model where the "correct" grouping to be discovered is well-defined, and use synthetic data to evaluate and compare three regression-based methods, namely, two proposed previously and one we introduce. We find that regression-based methods can recover the groupings even when the function is manifestly nonlinear; that multi-group methods offer an advantage over a single-group EQO; and crucially, that sim-pler (linear) methods can outperform more complex ones.

## Author summary

Natural microbial communities are highly complex, making predictive modeling difficult. One appealing approach is to make their description less detailed, rendering modeling more tractable while hopefully still retaining some predictive power. The Tree of Life nat-urally provides one possible method for building coarser descriptions (instead of

Inc.) reproducing all figures and simulations from scratch is included as S1 File.

**Funding:** This work was supported in part by the National Science Foundation grant PHY-2340791 to MT and PHY-2310746 to MT and OXC, as well as grants No. PHY-2309135 and the Gordon and Betty Moore Foundation Grant No. 2919.02 to the Kavli Institute for Theoretical Physics (KITP). The funders had no role in study design, data collection and analysis, decision to publish, or preparation of the manuscript.

**Competing interests:** The authors have declared that no competing interests exist.

thousands of strains, we could think about hundreds of species; or dozens of families). However, it is known that useful descriptions need not be taxonomically coherent, as illustrated, for example, by the so-called functional guilds. This prompted the development of computational methods seeking to propose candidate groupings in a data-driven manner. In this computational study, we examine one class of such methods, recently proposed in the microbial context. Quantitatively testing their performance can be difficult, as the answer they "should" recover is often unknown. Here, we overcome this difficulty by testing these methods on synthetic data from a model where the ground truth is known by construction. Curiously, we demonstrate that simpler approaches, rather than suffering from this simplicity, can in fact be more robust.

## Introduction

Microbial communities play key roles in global climate [1–3], food safety [4–6], and human health [7–10], but are highly complex [10–15]. To tackle this complexity, a key goal in ecology has been to derive methods of coarsening, e.g., functional groups or guilds [16, 17]. Such coarsened representations can be more reproducible than the microscopic characterization while still being predictive of properties of interest [17–23].

Over the years, multiple network-based algorithms for identifying biologically meaningful groups of organisms have been proposed [24, 25]. However, such approaches typically require extensive knowledge of species-species interactions, which is usually unavailable in microbial communities with a large number of species. Recently, Shan *et al.* [26] demonstrated the promise of a surprisingly simple methodology, ensemble quotient optimization (EQO), which can identify a "functional group" with respect to a specified property of interest (which we will call "function" for simplicity). For a continuous-valued function, the EQO algorithm is equivalent to a Boolean least square regression seeking to identify a subset of species whose combined abundance best correlates with the function while keeping the number of species in the group as small as possible (see Materials and methods, EQO for details). Moran et al. [27] used an approach that can be seen as a multi-group generalization of EQO. Compared to most modern applications of machine learning, EQO requires very little data. Further, in contrast to other methods of functional group identification, it only requires species abundances and the value of the function as input. This simplicity makes EQO highly appealing for microbial applications where such data is comparatively easy to collect. However, its empirical success is somewhat puzzling, as it amounts to modeling ecological function with a simple linear regression.

Realizing the promise of such methodology, and improving on its performance, requires understanding when and why EQO-like methods can succeed. Currently, validating the ability of such methods to discover biologically or mechanistically meaningful groups remains an open question. Of the three examples used in Shan *et al.* [26], only one (the data from [22]) had an independently established grouping against which the output could be compared (previously investigated in Ref. [28]). This issue is more general [24, 25, 29]. Empirical validation of grouping methods often relies on researchers' intuition, evaluating whether the groups "make biological sense." Such intuition-based validation can be compelling (e.g., Shan *et al.* [26] found that, when applied to the TARA Oceans [19, 30] dataset with nitrate as the observable of interest, EQO appropriately grouped aerobic and anaerobic ammonia oxidizers). However, to systematically compare or improve such methods, a quantitative assessment of their performance is required. This requires a context with a known "ground truth," against which the algorithm output can be compared.

Doing this in empirical datasets is difficult. Few empirical examples allow for the unambiguous delineation of the "true" functional groups [16, 29]; as a result, assessing the quality of a grouping is often qualitative and subjective. Here, we circumvent this limitation by adopting a model-based approach, evaluating algorithm performance on synthetic data from a model where the correct answer is, by construction, unambiguous. Of course, extrapolating model-based validation to applicability to real datasets requires caution. Such analysis can nevertheless provide useful insight in comparing algorithms and identifying their limitations. After all, an algorithm that fails to perform in the "clean" world of a model is unlikely to succeed in real life.

Specifically, we use a resource competition model with species catalyzing one of the steps of a degradation pathway with a specified topology. We take one of the degradation products (e.g. the final product) as the only quantity being measured (the "property of interest"). By construction, our model defines a unique "correct" grouping of species, namely, the grouping by reaction step performed. We use synthetic data from this model to compare the performance of three grouping algorithms: the single-group EQO of Ref. [26]; its multi-group generalization [27]; and a new algorithm we propose here, based on a Metropolis-like [31] search of the space of candidate groupings of species.

We find that, first, these algorithms can recover the expected groupings even when the function is manifestly nonlinear. Next, we show that multi-group methods can offer an advantage over the single-group EQO and, under some conditions, can correctly recover not only the group contributing to the function directly (in our model, the species producing the metabolite of interest), but also some information about the upstream groups whose influence is indirect. Finally, we present results indicating that on limited-size datasets with moderate measurement noise, simpler (linear) methods can outperform more complex ones.

## Results

### A consumer-resource model and the three methods for identifying groups

To evaluate the regression-based methods in a simplest model setting, we adopt a chemostat consumer-resource model with cross-feeding, where the metabolism is designed so that there is an evident way to group species. The model includes $S$ microbial species whose abundances are denoted by $n_\mu$ ($\mu = 1, \ldots, S$) and $N$ metabolites whose concentrations are denoted by $m_i$ ($i = 1, \ldots, N$). These $N$ metabolites are designed to form a linear degradation chain $1 \rightarrow 2 \rightarrow \cdots \rightarrow N$. The linear pathway topology is a convenient place to start, since intuitively, it is one where a linear-regression-like approach is most likely to succeed. More complex pathway topologies, and various other ways to challenge the approach, will be discussed later. Species are designed to catalyze at most one reaction of the chain, which naturally classifies them into $N$ groups (Fig 1A). Specifically, species in group $i$ ($i \leq N - 1$) feed on metabolite $i$ and transfer a fraction $w_i$ of it into resource $i + 1$, while species in group $N$ are not involved in the chain. The concentration of the end product $m_N$ is taken as the function of interest. We termed the last group in the chain (group $N - 1$) "the group of direct producers".

Besides these $N$ metabolites $m_i$, we assume there are $H$ other generalised depletable resources for species to exploit, which create additional variability and competition; all resources are assumed to be substitutable for simplicity [32–34]. The availability $h_a$ ($a = 1, \ldots, H$) of generalised resource $a$ is assumed to be a monotonically decreasing function of the total exploitation; for simplicity, we follow Ref. [23] and assume this dependence to take a simple

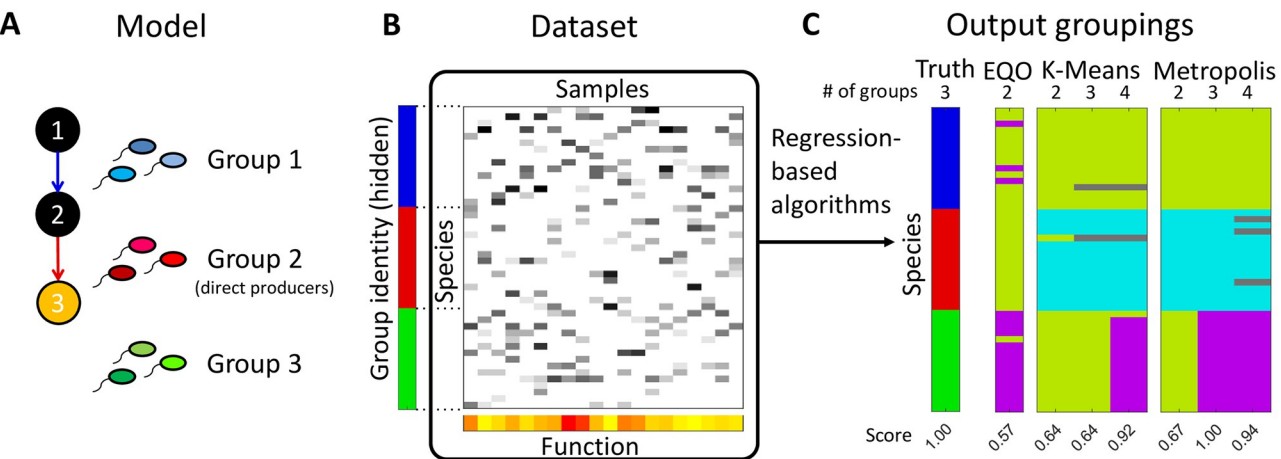

**Fig 1. Using synthetic data to test group-searching algorithms in a context where the correct grouping of species is known and uniquely defined.**
(A) We adopt a resource competition model with cross-feeding. The reaction network is assumed to form a linear degradation chain $1 \rightarrow 2 \rightarrow \cdots \rightarrow N$ with the end-product concentration (metabolite $N$, orange) taken as the function of interest (shown with $N = 3$ as an example). Species can perform at most one reaction of the linear chain, which naturally groups them into $N$ groups ($N - 1$ groups consuming metabolite $1, \ldots, N - 1$, and a group not involved in the chain). The model also includes $H$ other resources for species to compete over, which create additional variability (omitted for clarity, see text). (B) The synthetic dataset is generated by repeatedly selecting a random subset of 15 species and allowing the community to equilibrate (see Eqs 1a–1d). The final abundances and function (concentration of resource $N$) are corrupted with Gaussian noise of relative strength $\epsilon$ emulating "measurement noise," and the resulting values are recorded as a "sample" in the dataset. (C) We use the synthetic data as input for three families of regression-based algorithms: the EQO of Ref. [26] (which groups species into two groups), and two families we call K-means and Metropolis (see text), which can return any specified number of groups. The panel shows representative outputs of these algorithms for $N = 3$ metabolites and for the number of groups indicated at the top. Species assigned to the same group are shown in the same color. Outputs are quantitatively scored (see text) based on the similarity to the "ground-truth" grouping hard-coded into the model (left-most row). Higher score is better; a score of 1 corresponds to a perfect matching.

hyperbolic form. Putting this together, the dynamics is described by the following equations:

$$\frac{dn_{\mu}}{dt} = n_{\mu}\left[\sum_i (1 - w_i)\tau_{\mu i}m_i + \sum_a \sigma_{\mu a}h_a - \chi_{\mu}\right] \tag{1a}$$

$$\frac{dm_1}{dt} = R_1 - m_1\sum_{\mu}\tau_{\mu,1}n_{\mu} - d_1 m_1 \tag{1b}$$

$$\frac{dm_i}{dt} = w_{i-1}m_{i-1}\sum_{\mu}\tau_{\mu,i-1}n_{\mu} - m_i\sum_{\mu}\tau_{\mu i}n_{\mu} - d_i m_i \qquad (\text{for } i > 1) \tag{1c}$$

$$h_a = h_a(\{\sigma_{va}\}, \{n_v\}) = \frac{h_0^a}{1 + \sum_v \sigma_{va}n_v/K_a}. \tag{1d}$$

Of the metabolites in the chain, only the first is supplied externally (at rate $R_1$); for $i > 1$, the only source of metabolite $m_i$ is secretion by species consuming metabolite $m_{i-1}$. Thus, the function (concentration of $m_N$) is naturally nonlinear: producing the final product requires all $N - 1$ groups to be present. The quantity $w_i$ is the transfer ratio of each reaction; $d_i$ is the decay rate of metabolite $i$. The generalized resources are described by parameters $h_0^a$ (the highest benefit the resource can provide) and $K_a$ (the exploitation level at which this benefit is depleted by half).

A species $\mu$ is defined by its role in the metabolic chain ($\tau_{\mu i} \in \{0, 1\}$ equals 1 if the species can consume metabolite $i$ and 0 otherwise), its utilization strategy of other generalized resources ($\sigma_{\mu a} \in \{0, 1\}$ equals 1 if it can exploit resource $a$ and 0 otherwise), and its maintenance cost $\chi_\mu$. Here, we pick

$$\chi_\mu = \sum_i (1 - w_i)\tau_{\mu i} + \sum_a \sigma_{\mu a} + \varepsilon x_\mu, \qquad (2)$$

where $\varepsilon$ is a small quantity (taken to be 0.01 in this paper) and $x_\mu$ is a Gaussian random number with mean zero and variance one. This choice follows the convention of Ref. [32], so that species able to benefit from more resources also have a larger cost, and neither generalists nor specialists are obviously favored. (At equilibirum, we expect $m_i \approx 1$, $h_a \approx 1$. Eq (2) sets the cost $\chi_\mu$ to approximately match the expected benefit, whatever the species' strategy. As a result, the winners and losers of the competition are determined by the luck of the draw of the small random contribution $x_\mu$).

The model makes many simplifications (perfect conversion efficiency, substitutable resources, ignoring Liebig's law. . .) adopted for simplicity, following previous work [32–34] to minimize the number of model parameters. However, for our purposes, two assumptions are especially worth highlighting. The binary $\tau_{\mu i}$ correspond to species that are contributing to at most one reaction of the chain (no promiscuity), making the grouping unambiguous. Within each group, species differ in their utilization of the generalized resources, but the contributions to the reaction of interest are assumed to be the same (no heterogeneity). The role of these two assumptions will be examined shortly.

Species abundances determine the reaction fluxes and thus the value of the functional property of interest $m_N$ (the concentration of metabolite $N$). With the model defined above, it can be shown (see S1 Text Section 1) that at equilibrium,

$$m_N = \frac{R_1}{k_N}\frac{w_1 T_1}{T_1 + d_1}\frac{w_2 T_2}{T_2 + d_2}\cdots\frac{w_{N-1} T_{N-1}}{T_{N-1} + d_N}, \qquad (3)$$

where $T_i = \sum_\mu \tau_{\mu i} n_\mu$ is the total abundance of the functional group $i \in \{1, \ldots, N - 1\}$. Thus, the individual species $n_\mu$ affect the value of the function only via the combined group abundances $T_i$, but the relationship between function and group abundances is manifestly nonlinear in this model.

In this paper, we set the parameters as follows. The metabolite transfer ratios $w_i$ and the decay rates $d_i$ are the same for all $i$: $w_i \equiv w = 0.5$ and $d_i \equiv d = 1$. The supply rate $R_1$ is set to $R_1 = \prod_{i=1}^{N-1} w_i^{-1} = w^{-(N-1)}$, compensating for the losses at each reaction to ensure that, if we change $N$, the value of the function $m_N$ remains of the same order. Finally, the $H$ generalized resources are selected to be identical for simplicity, with $h_0^a \equiv h_0 = 3$ and $K_a \equiv K = 1$ for all $a$.

We generate the synthetic dataset by first generating a species pool (i.e., generating $\{\sigma_{\mu a}\}$ and $\{m_\mu\}$, see Materials and methods), and then repeatedly selecting a random subset of species and allowing the community to equilibrate according to Eqs 1a–1d. The final species abundances $\{n_\mu\}$ and function $m_N$ are corrupted with Gaussian noise of relative strength $\epsilon$ emulating "measurement noise", and the resulting values are recorded as a "sample" in the dataset (Fig 1B).

We use the synthetic data as input for three families of regression-based algorithms (see Materials and methods for details). The first is the EQO proposed by [26] which we modified to incorporate the Akaike Information Criterion (AIC) into the optimization process. In the second method, the coefficients of a multiple linear regression against all species are fed into K-Means clustering for grouping [27]. We term this method "K-Means," for simplicity. The

third is a new algorithm we propose. In this approach, the root-mean-square-error (RMSE) of a multiple linear regression against (candidate) group abundances takes the role of energy, which we seek to minimize while searching the coarse-graining space with a Metropolis-like [31] algorithm. We term this algorithm "Metropolis." All three algorithms are linear-regression-based (but the third can be extended to include higher-order terms; we will return to this point later). By design, EQO always outputs two groups ('functional' and 'non-functional' species); in contrast, K-Means and Metropolis can return any specified number of groups. Representative examples of the output groupings of each algorithm (with the ground truth containing $N = 3$ groups) are shown in Fig 1C.

To evaluate the quality of such groupings, we use a metric based on Jaccard Similarity. First, we define the "recovery quality" of a group in the ground truth as the Jaccard Similarity between this group and its best match in the grouping being evaluated. Then, the overall quality score of a grouping is defined as the average recovery quality of all the ground-truth groups (see S1 Text Section 2 for details). By construction, this score is between 0 and 1, where 1 indicates perfect matching. Perfect matching can only be expected when the number of groups in output ($k$) equals the number of groups in the ground truth ($N$). If $k < N$, then the highest possible score is $k/N$, which we call the performance ceiling of a $k$-group output for $k \leq N$ (see S1 Text Section 2). The quality scores for each of the example groupings in Fig 1C are shown below them.

## Linear-regression-based algorithms perform well, with multi-group algorithms recovering more information

We begin by evaluating the three algorithms on synthetic datasets with $N = 3$ true groups of 16 species each, for a total of $S = 48$ species (groups 1 & 2 successively degrade metabolite 1 into metabolite 3, while group 3 is "nonfunctional"). For our first test, we consider the most favorable regime with a large number (900) of samples and low noise (10%). We follow the protocol of Fig 1 to test each of the algorithms on 50 synthetic datasets. The quality scores of all the 2- and 3-group outputs are summarized in Fig 2A and 2B. (The groupings themselves are shown in S4 Fig).

For 2-group outputs (Fig 2A), all three algorithms perform substantially better than random, with K-Means and Metropolis approaching the performance ceiling of 2-group groupings ($k/N = 2/3$, dashed line). As one might expect, in most cases, the groupings identified by the 2-group algorithms distinguish direct producers from the rest of the species (see S4 Fig). Note that while EQO groups species into two groups, it assumes that only one of them (the "functional group") affects the level of function. However, the "nonfunctional group" may also affect function through competition with functional species. This may help explain the comparatively low performance score of this algorithm: for our synthetic data, removing this restriction improves performance (see S1 Text Section 3).

For 3-group outputs, both multi-group algorithms cross the performance ceiling of 2-group methods (Fig 2B). Examining the output reveals that this is due to resolving not only the group of direct producers, but also (at least some of) the species that contribute to an upstream reaction (group 1); see S4 Fig. Thus, we confirm that multi-group algorithms can recover more information on the community structure.

The analysis just described was performed for a particular dataset size and noise magnitude. The effect of these parameters is presented in Fig 2C, which shows the average score (over 50 synthetic datasets) of the 3-group Metropolis. As expected, the difficulty of the task increases if the dataset is small and/or noisy. One also expects the method to perform less well if the generalized resources are made to have a larger impact on species growth rates; see S1 Text Section

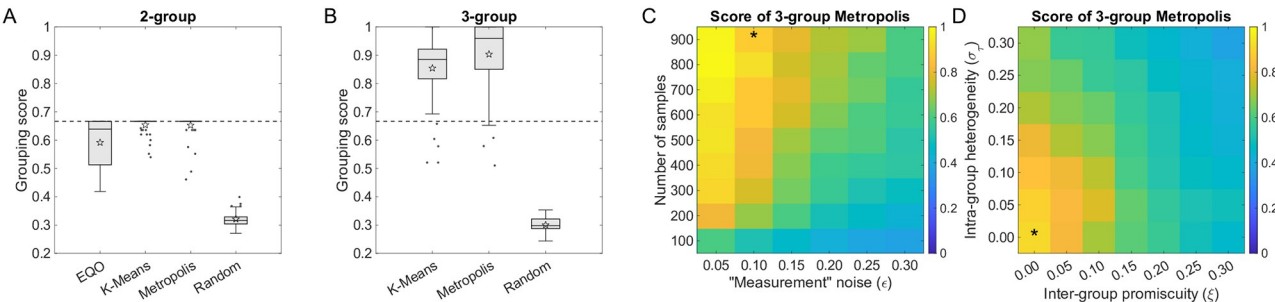

**Fig 2. Linear-regression-based algorithms succeed at identifying the correct functional groups in synthetic data, and multi-group algorithms recover more information.** (A), (B) Algorithm performance, evaluated over 50 simulated datasets generated as described in Fig 1 with $N = 3$ true groups, 900 samples and 10% simulated measurement noise. Performance scores (i.e., the similarity of result to the 3-group ground truth) are shown as box plots, separately for the 2-group outputs of all three algorithms (A) and for the 3-group outputs of the K-Means and Metropolis methods (B). Performance scores of random groupings are shown as controls. Boxes represent the interquartile range (IQR) between the first and third quartiles; the line inside represents the median. Whiskers show the lowest and highest values within $1.5 \times IQR$ from the first and third quartiles, respectively. Points that fall outside of the range of the whiskers (the outliers) are shown explicitly. Stars mark the mean values. The dashed horizontal line is the theoretical performance ceiling of any 2-group grouping when evaluated against the 3-group ground truth (compare to Fig 1C); 3-group methods can cross this bound. All three algorithms perform substantially better than random, with Metropolis scoring the highest. (C) Heatmap shows the performance of the 3-group Metropolis as a function of the measurement noise magnitude and the number of samples in the dataset. (D) Heatmap of the performance of the 3-group Metropolis under increasing intra-group heterogeneity ($\sigma_\tau$) and inter-group promiscuity ($\xi$) of species, to show the limitation of linear-regression-based algorithms under fuzzy ground truth groupings. In (C) and (D), each pixel is an average over 50 synthetic datasets. The star indicates the parameters used in (A) and (B).

3. Here and below, we focus on the Metropolis algorithm for clarity of presentation, as it appears to perform best, at least on the synthetic data used here. The scores for 2-group outputs and for the other two algorithms behave similarly, and are presented in S5 Fig.

To further challenge the algorithms to detect their limitation, we tweak the model in two ways, relaxing some of the assumptions to make the ground truth grouping less clear. First, we allow species in the same group to vary in their contribution to the respective degradation reaction. Specifically, instead of setting all non-zero terms of $\tau_{\mu i}$ to be the same, we draw them from a distribution with width $\sigma_\tau$; we call this intra-group heterogeneity. Second, we consider species that are increasingly promiscuous rather than specializing in a single reaction step (in other words, we let them have a small rate $\xi$ for reaction(s) not belonging to its group); we call this inter-group promiscuity. The details are described in Materials and methods. Fig 2D presents the heatmap of Metropolis performance as a function of the heterogeneity and promiscuity parameters (see also S6 Fig). We see that the algorithm can tolerate some deviations in either direction; however, for high heterogeneity or promiscuity when the group identity becomes increasingly fuzzy, performance begins to fall and approaches the score of a random grouping.

## Groups affecting the function more directly are easier to recover

The overall quality score defined above and analyzed in Fig 2 is a summary statistic averaged over all the groups in the output. However, some groups may be recovered better than others. To characterize this, we now increase the length of the degradation chain and focus on the recovery quality of individual groups, measured by the Jaccard Similarity between a given true group and its closest match in the algorithm output. To make the results otherwise comparable as we increase the length of the chain, the total number of species is kept as similar as possible under the constraint that each group contains the same number of species (see Materials and methods for details). Throughout this analysis, the dataset size is held steady at 900 samples and the noise magnitude is kept at 10%.

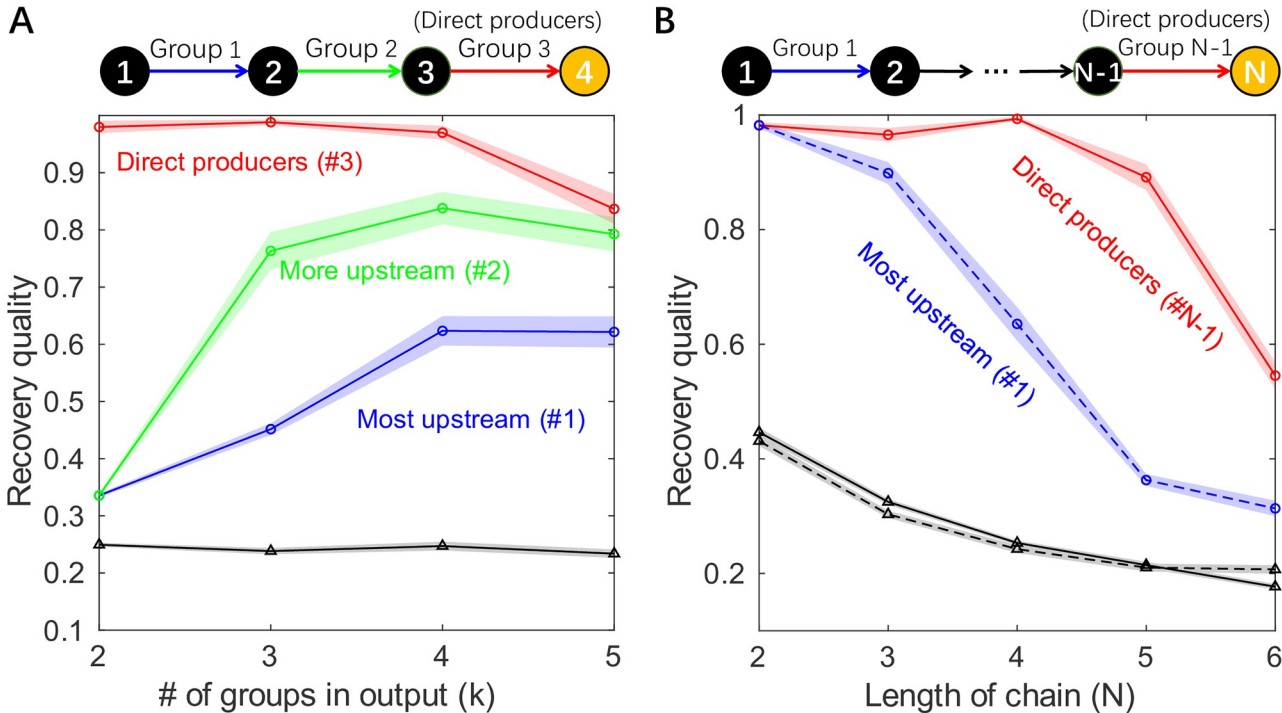

**Fig 3. Identifying the groups becomes harder when the degradation chain is long, especially for groups catalyzing upstream reactions.** (A) The panel shows the ability of the Metropolis algorithm to recover the true functional groups within a linear degradation chain with $N = 4$ metabolites. The recovery quality of a group is defined as the Jaccard Similarity between the true functional group and its closest match in the algorithm output. Here, the recovery qualities of each of the true functional groups (groups 1, 2, and 3) are shown as a function of $k$, the number of groups requested from the algorithm. Recovery qualities attained by random $k$-group groupings are shown as controls (black line with triangle markers). As the number of groups $k$ in the output increases, the algorithm first finds group 3 (direct producers), then group 2 and group 1, with ever-decreasing recovery quality. (B) The recovery quality of Metropolis of the most upstream group (group 1, blue dashed line) and the direct producers (group $N - 1$, red solid line) in the $N$-metabolites degradation chain, shown as a function of $N$. Recovery quality of group 1 is reported for $N$-group algorithm outputs while that of direct producers is for 2-group outputs (see text.) As controls, the recovery qualities (of arbitrary group) by $N$-group and 2-group random groupings are shown as black dashed line and black solid line, respectively. As the chain becomes longer, the ability to recover the most upstream group drops quickly, whereas the direct producers are adequately recovered up to length 5. In both panels, shading indicates the standard error of the mean over 50 synthetic datasets, circles indicate data points for Metropolis while triangles for random groupings.

Fig 3A shows the recovery quality of each of the groups in a degradation chain of length $N = 4$, as identified by the Metropolis algorithm for various $k$. Similarly to the results of the previous section, at $k = 2$ (two-group output), the group of direct producers is the only group recovered. Increasing $k$ makes it possible to resolve other groups, but the recovery quality drops as groups get further away in the chain.

To further illustrate this point, Fig 3B compares the recovery quality of direct producers (group $N - 1$) and the most distant upstream group (group 1), as a function of the length of the chain. (Note that identifying the most distant group requires using $k = N$, while the direct producers are best identified by setting $k = 2$; see Fig 3A). We see that as the chain becomes longer, the ability to recover the most distant group drops quickly, whereas the direct producers are adequately recovered (in this example) up to length 5. These results quantitatively confirm the intuition that groups of species affecting the function more directly are easier to recover, while further illustrating the ability of multi-group algorithms to recover more information on community structure.

While the ability to recover upstream groups is remarkable, we hypothesized that it is facilitated by our choice of a particularly simple (linear) topology of the degradation pathway. To

test this, we considered several other reaction topologies, as well as other choices for the quantity of interest beyond the case of the end-product metabolite of a linear degradation chain. Specifically, we let the function be an intermediate product of a linear degradation chain; one of the end products in a degradation chain with a branch; or the common end product of two linear chains. This analysis is presented in the S1 Text Section 4. In the first two cases, Metropolis can again identify all the functional groups, while in the last, it can only recover the groups which directly produce the metabolite of interest. In summary, our analysis confirms that for a function associated with multiple groups, the group which affects (correlates with) the function the most will in general be easiest—and sometimes the only one—to be found.

## Finding the right variables can be easier than finding the right model

Given the promising performance of linear-regression-based algorithms, it is natural to ask whether algorithms based on more complex models could do better. Of note, our Metropolis algorithm can be generalized to any model of community function that can accept the combined group abundances as input, and return the RMSE of the prediction. Thus, the Metropolis algorithm can be used to test different models under the same framework. Here we consider a generalization to a regression with both linear and quadratic terms, which we term the 'quadratic Metropolis.' To emphasize this difference, the original version considered above will from now on be referred to as 'linear Metropolis.'

To compare these two versions, we evaluate them on the same synthetic datasets with $N = 3$ true groups as in Fig 2. Before comparing their performance, we note that, by construction, each algorithm constructs *two* objects. First, it returns a set of coarsened *variables*—i.e., the groups. Second, it also identifies a predictive *model* that uses these variables to predict the function (see Eqs (4a) and (4b) in Materials and methods)—in our case, the specific instance of the linear or quadratic regression model. When comparing the performance of the linear and quadratic versions of the algorithm, it is important to be clear that in this work, our primary focus is on identifying the variables. In contrast, the prediction error of the model is only a means to an end: we assume, or hope, that the regression model trained on the correct variables will have a lower RMSE than a model trained on the wrong variables.

Intuitively, the quadratic model should predict the function better since it has more parameters, and the true structure-function mapping (Eq (3)) is nonlinear. This is indeed the case, as demonstrated in Fig 4A which shows the difference in out-of-sample $R^2$ (Eq (5), linear minus quadratic, averaged over 100 datasets) as a function of the number of samples and noise magnitude. (See Materials and methods for details). In some of the parameter range, the quadratic model has higher predictive power. Crucially, however, finding the right *variables* is distinct from finding the right *model*. The heatmap in Fig 4B uses the same data as panel A, but plots the difference of grouping quality scores identified by the two algorithms. Putting these two panels together, we distinguish three regimes, as indicated by dashed lines. In the first regime (many samples, low noise), the quadratic model is better at both predicting the function and detecting groups. In the second, the linear version is better at identifying variables, even though the quadratic is better at predicting the function. In this regime, the higher expressivity of the more complex model appears to hinder the algorithm's ability to correctly identify the variables. Finally, in the third (and arguably the most relevant) regime of few samples and high noise, the quadratic version, somewhat surprisingly, performs worse at both tasks. This is because at some point, the failure to identify the variables also limits its ability to predict the function. (Of course, one caveat is that in this region of parameter space, the task is especially hard: even for the better-performing linear method, the absolute quality of group prediction remains relatively poor; see Fig 2C).

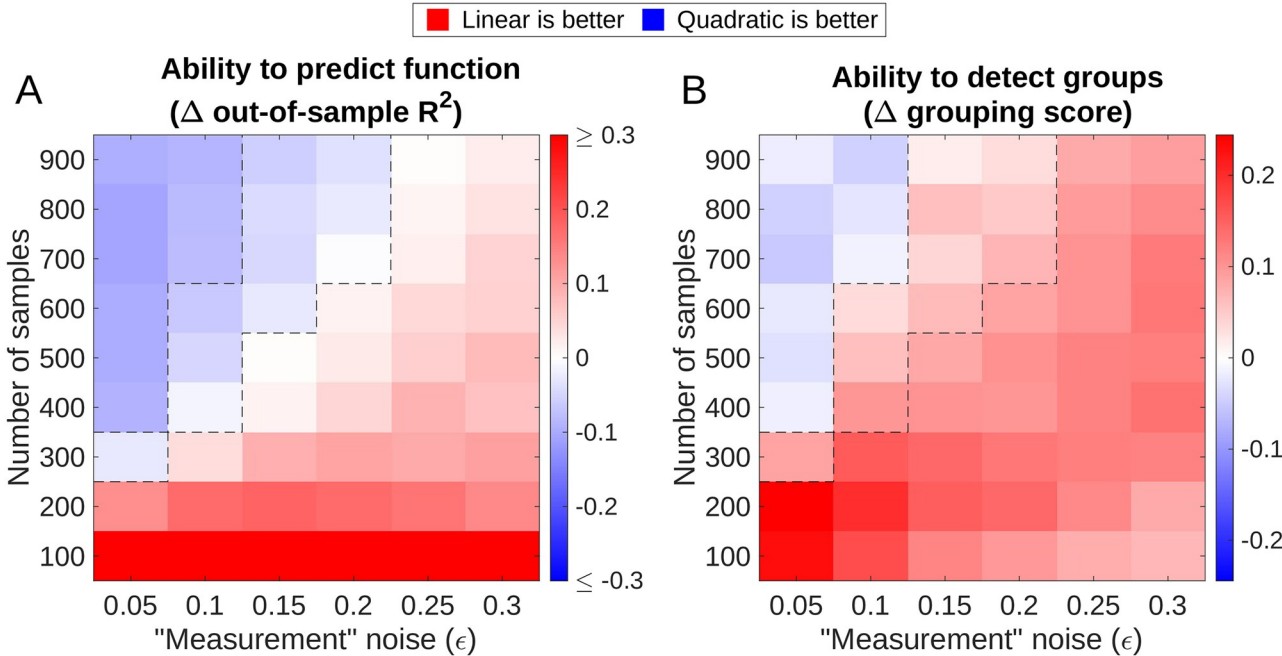

**Fig 4. If datasets are small and/or noisy, linear-regression-based algorithms for identifying functional groups outperform more complex versions.** We compare the performance of the linear-regression-based Metropolis algorithm to a more expressive version that includes quadratic terms. Both versions are evaluated on the same synthetic datasets with a 3-group ground truth. Each algorithm return a set of coarsened *variables* (a grouping of species into three groups) and a *model* that uses these variables to predict the function. (A) The model identified by the quadratic Metropolis is often more predictive of the function (blue). The heatmap shows the difference in out-of-sample coefficient of determination ($R^2$). More specifically, we plot the $R^2$ of the best linear model minus the $R^2$ of the best quadratic, where "best" refers to the model identified by the corresponding Metropolis algorithm over its finite runtime (10000 steps). (B) Nevertheless, even when the linear algorithm loses in $R^2$, the grouping it identifies can be a better representation of the underlying ground truth. The heatmap shows the difference in the quality score of the grouping (linear minus quadratic). The panels highlight that the task of identifying a predictive coarsening of an ecosystem (B) is distinct from the task of predicting the function well (A), and for small or noisy datasets, the former is best accomplished by a simpler method. Each pixel is an average over 50 datasets. Dashed lines mark the boundaries between the three regimes discussed in the main text.

In conclusion, we find that for small or noisy datasets, the task of identifying a predictive coarsening of an ecosystem ("finding the right variables") can be easier than the task of predicting the function well ("finding the right model"), in the precise sense that—at least in the example considered here—it is best accomplished by a simpler method.

## Discussion

In this work, we examined the ability of several simple algorithms to recover meaningful "functional groups" of microbial taxa using only the information on species abundances and a single function of interest across a collection of samples. For this, we used synthetic data generated from a model for which the most sensible grouping could be defined unambiguously. This allowed us to quantitatively assess an algorithm's performance by comparing its output against the expected "ground truth". We found that, first, simple regression-based methods could indeed correctly recover a substantial amount of information about the underlying community structure, at least in the simplest scenarios considered in our model. Second, we showed that multi-group algorithms can offer an advantage over the two-group EQO proposed previously. Finally, and most importantly, our analysis indicates that under some conditions, particularly for datasets that are small and/or noisy, simpler (linear) methods can outperform more complex ones.

Our minimal model included many simplifications, and considered only the simplest reaction topologies. Even in these favorable cases, we have seen that realistic details, such as inhomogeneity of species contributions to function, reduce performance. Other complications could degrade performance further; for example, the method is unlikely to succeed for functions with non-monotonic dependence on group abundances, or instances when individual species' contributions are strongly context-dependent.

Of the three algorithms evaluated here, the newly proposed Metropolis algorithm performed best. However, it is clear that the findings of a model-based evaluation such as ours should be interpreted with caution. Whether the Metropolis-based algorithm would retain this relative advantage in real-world applications remains to be established.

## Materials and methods

### Linear-regression-based algorithms

In this work, we test three linear-regression-based algorithms, termed "EQO," "K-Means," and "Metropolis." For all three algorithms, the input is an abundance table (matrix) $A_{a\mu}$ and a column vector $Y_a$ of the values of the functional property in each sample (here and below, row index $a$ labels samples, column index $\mu$ labels species). The output is a grouping of species into $k$ groups for one or several $k$. This section presents the technical details of these algorithms. The algorithm for generating random $k$-group groupings with given $k$, which serves as control, is also presented here.

**EQO.** The EQO algorithm was proposed by Ref. [26]. For a continuous function, this algorithm is equivalent to a Boolean least square regression which selects from the community a subset of species (the "functional group") whose combined abundance correlates with the function of interest [26]. As such, it constructs a 2-group grouping of species: those included in the functional group, and those that are not.

Each candidate grouping can be represented by a column Boolean vector $\vec{x} \equiv \{x_\mu\}$ of length $S$ (the total number of species), where the species included / not included in the functional group are encoded by setting $x_\mu = 1$ and $x_\mu = 0$, respectively. The EQO algorithm executes a search in the space of such Boolean vectors. For a given candidate $\vec{x}$, we first calculate the abundance of the functional group (in each sample) $f_a = \Sigma_\mu A_{a\mu} x_\mu$, then perform a 1-dimensional linear regression (with an intercept), with $Y_a$ as response and $f_a$ as predictors. We then calculate the Akaike Information Criterion (AIC) of this regression as AIC $= 2\kappa + n \log(RSS/n)$, where $\kappa$ is the size of the functional group (the number of nonzero components in $\vec{x}$) and $n$ is the number of samples. RSS is the residual sum of squares of the regression. Thus, with $A_{a\mu}$ and $Y_a$ given, AIC is a function of $\vec{x}$. We use the MATLAB built-in function `ga` to find the optimal $\vec{x}$ which minimizes the function AIC($\vec{x}$) using a genetic algorithm. We set the options of `ga` as follows: `FunctionTolerance=1e-9, MaxStallGenerations=500, MaxGenerations=10000` and `PopulationSize=100`.

Of note, this implementation is slightly different from the protocol of Ref. [26], where RSS is first minimized for a range of (fixed) $\kappa$ values, after which the AIC is calculated and the lowest value is selected. For our application, we found that the constraint of holding $\kappa$ fixed slows down optimization significantly, so we chose to combine the two successive steps into a single optimization process.

Note that although EQO groups species into two groups, only one of them is assumed to affect the function. To explore the effect of this assumption, for our testing we also considered a variant of EQO where this constraint is relaxed (EQO-2g; see S1 Text Section 3). For our synthetic data, we found that removing this restriction improves performance.

**K-Means.** In this method, one first performs an $S$-dimensional linear regression (with intercept), using the function $Y_a$ as response and the abundances of each species $A_{a\mu}$ as predictors. Then the coefficients of all species are fed into the K-Means clustering algorithm (performed by the MATLAB built-in function `kmeans`) which groups the coefficients (and thus species) into $k$ groups for the specified $k < S$. This heuristic approach is very naive, to the point that it is rather surprising it can be as successful as it is (cf. Fig 2). When it does succeed, it offers the advantage of being incomparably faster than either of the other methods.

**Metropolis.** The aim of this algorithm is to find a set of $k_{max}$ optimal groupings $\mathcal{P}^* = \{\mathcal{P}_1^*, \mathcal{P}_2^*, \ldots, \mathcal{P}_{k_{max}}^*\}$ where $\mathcal{P}_k^*$ is the optimal $k$-group grouping which gives the lowest RMSE through a linear regression. The approach, briefly, is to keep in memory a list of best current candidates $\mathcal{P}$ and the associated RMS error values $\mathcal{E} = \{\mathcal{E}_k\}$. We then perform $M$ steps trying new groupings (by splitting or merging groups of the groupings already in $\mathcal{P}$), updating the list as better groupings are found, and then assume the candidates are good enough, setting $\mathcal{P}^* = \mathcal{P}$.

More specifically, the algorithm proceeds as follows:

1. **Initialization**. The list of candidates $\mathcal{P} = \{\mathcal{P}_1, \mathcal{P}_2, \ldots, \mathcal{P}_{k_{max}}\}$ is initialized by randomly generating a series of $k$-group groupings $\mathcal{P}_k$ (see Section "Random grouping" below). For each $\mathcal{P}_k$, we then calculate the combined abundance of each group and perform a $k$-dimensional linear regression (with intercept) with function $Y_a$ as response and group abundances as predictors. The RMSE of this regression is recorded as $\mathcal{E}_k$.

2. **Main loop**

   a. **Construct a new candidate:** Randomly choose one of the groupings $\mathcal{P}_k$ from the current list $\mathcal{P}$. If $1 < k < k_{max}$, randomly split one of the groups in $\mathcal{P}_k$ into two (with probability $p = 0.5$), or randomly merge two groups into one (with probability $1 - p$), thus obtaining a new grouping $P_{k'}$ with a different number of groups $k' \neq k$. If $k = 1$ or $k = k_{max}$, only one of these operations is possible (respectively, only splitting or only merging), and is performed with probability 1.

   b. **Evaluate the new candidate:** Calculate the combined abundance of each group in the new grouping $P_{k'}$; perform a $k'$-dimensional linear regression (with intercept) with function $Y_a$ as response and group abundances as predictors; and record its RMSE as $E_{k'}$.

   c. **Update the $\mathcal{P}$ list:** Compare $E_{k'}$ to $\mathcal{E}_{k'}$ (the RMSE of the $k'$-group grouping $\mathcal{P}_{k'}$ currently stored in $\mathcal{P}$). With probability $\min\{\exp(-\beta(E_{k'} - \mathcal{E}_{k'})), 1\}$, replace the currently stored grouping with $P_{k'}$.

3. Repeat the main loop $M = 10000$ times, then return the current list of candidate groupings $\mathcal{P}$ as the best guess of the optimal list $\mathcal{P}^*$.

In practice, we found $\beta = \infty$ to perform best, so we set $\beta = \infty$ for all the tests in this paper. This "zero-temperature" regime is usually undesirable, as it can cause optimization to become stuck in a local optimum. However, for our application, we have not observed this to occur: our approach of storing the list of candidate groupings for all $k$ always maintained a large number of accessible moves.

Empirically, we found that a sufficiently large $k_{max}$ is required for a good performance, even if we are ultimately only interested in output groupings with small $k$. Throughout our analysis, we set $k_{max} = 20$. With this $k_{max}$, $\beta = \infty$ was always found to perform best in our testing.

For the quadratic Metropolis introduced in Fig 4, we replace the linear regression with a regression with both linear terms and quadratic terms (which includes both $T_i^2$ terms and the

$T_iT_j$ cross-product terms, with $T_i$ the combined abundance of group $i$), as well as an intercept like before. Everything else is identical for both versions. Note that the number of coefficients in this regression model is quadratic in $k$ (the number of groups), not $S$ (the number of species). Thus, for all the figures shown, the model coefficients were well-constrained even with the lowest dataset size assayed (100 samples).

**Random grouping.** We generate a random $k$-group grouping of all $S$ species as follows. First, randomly permute the $S$ species (represented by $S$ integers from 1 to $S$). If we think of this reordered set as a list of integers, with $S - 1$ "gaps" between them, then selecting a random partitioning into $k$ non-empty groups is equivalent to randomly selecting $k - 1$ of these "gaps" as the locations of group boundaries.

## Simulation details

To generate the datasets, we first generate a pool of $S$ species, which means randomly generating the matrix $\{\sigma_{\mu a}\}$ as a sparse binary matrix with density 0.3 (i.e., each entry $\sigma_{\mu a}$ is independently set to 1 with probability 0.3, and 0 otherwise) and generating the maintenance cost $\chi_\mu$ of each species according to Eq 2. The number of generalised resources $H$ is set to be 15 except in panel D of S2 Fig, where it is set to 30. The total number of species $S$ is set to 48, with each group containing $S/N$ species. The one exception is the $N = 5$ case of Fig 3: since 48 is not divisible by 5, we instead set $S = 50$ (so each group contains 10 species). Other parameters have been stated in the main text.

For a dataset consisting of $n$ samples, for each sample we randomly select 15 out of all $S$ species, whose initial abundances are set to 1 (while those of the remaining species are set to 0). The initial values of all $m_i$'s are set to 1. We use the MATLAB built-in function `ode45` to simulate Eq (1) to approximate equilibrium. We then record the final abundances of all species $n_\mu$ and the concentration of the functional molecule $m_N$. We repeat this procedure $n$ times to obtain $n$ samples. We then multiply each element of the abundance table and function by an i.i.d. random variable drawn from a normal distribution with mean 1 and width $\epsilon$. Negative values are set to 0. The magnitude to $\epsilon$ tunes the strength of measurement noise.

## Intra-group heterogeneity and inter-group promiscuity

Here we describe the operation details of analysis in Fig 2D. As mentioned in the main text, originally the degradation rate $\tau_{\mu i}$ equals 0 or 1 with each species degrading at most one metabolite of the chain. We first add inter-group reaction promiscuity by replacing each zero $\tau_{\mu i}$ for $i \leq N - 1$ (the last metabolite cannot be degraded) with a small value $\xi$ so that each species is endowed with a small catalytic activity for all reactions, not just the one defining its group identity. We let $\xi$ take a series of values from 0 to 0.3 to model different degrees of promiscuity. We then add species heterogeneity within a group as follows: for each species we draw an i.i.d. random number $\eta_\mu$ from lognormal distribution with parameter $\mu_\tau = \ln 1$ fixed and $\sigma_\tau$ varying from 0 to 0.3 to model gradually increasing heterogeneity. We then scale all reaction rates $\tau_{\mu i}$ of a given species $\mu$ by $\eta_\mu$ as a global factor. This rescaling can be understood as the uncertainty in counting abundances in practice (e.g., due to the species carrying a different number of copies of the 16S RNA).

## Comparison of abilities to predict function of linear and quadratic Metropolis

Here we describe the detailed protocol of the comparison of abilities to predict function of linear and quadratic Metropolis shown in Fig 4A. As mentioned in the main text, besides the groups (*variables*) these versions of Metropolis also identify two *models* for predicting the

function:

$$\hat{Y}_L = b^L + \sum_i c_i^L T_i, \tag{4a}$$

$$\hat{Y}_Q = b^Q + \sum_i c_i^Q T_i + \sum_{i,j} d_{ij}^Q T_i T_j, \tag{4b}$$

where $T_i$ is the combined abundance of group $i$, and $\hat{Y}_L$ and $\hat{Y}_Q$ are the function predicted by linear and quadratic model, respectively. The two models are uniquely determined by their regression coefficients, $\{b^L, c_i^L\}$ for linear model and $\{b^Q, c_i^Q, d_{ij}^Q\}$ for quadratic model. In Fig 4A we are comparing the predictive power of these two models.

To do so, for each generated pool of species (see Simulation details), we now generate 2 datasets consisting of the same number of samples. One of them is used as the training set, while the other one is set aside as the testing set. The training set is fed into the two versions of Metropolis, which are now required to output not only the optimal groupings they find, but also the coefficients of the corresponding models (Eqs 4a & 4c) trained on the training set. (Specifically for Fig 4, we only ask for the 3-group grouping and its regression coefficients). We then test their abilities to predict the function in the testing set. The out-of-sample $R^2$ is calculated as

$$R^2_{out-of-sample} = 1 - \frac{\sum_a (Y_a - \hat{Y}_a)^2}{\sum_a (Y_a - \bar{Y})^2}, \tag{5}$$

where $\hat{Y}_a$ is the predicted value of function (for sample $a$), $Y_a$ is the true value, and $\bar{Y} = \frac{1}{N}\sum_a Y_a$ is the average of $Y_a$. The differences of out-of-sample $R^2$ of the two versions of Metropolis (linear minus quadratic) are shown in Fig 4A.

## Supporting information

**S1 File. Simulation Scripts And Data.** MATLAB simulation code and scripts generating Figs 2–4 and S2–S6 Figs from scratch, as well as data used for these figures shown in the current manuscript.
(ZIP)

**S1 Text. Supplementary information text.** Details of the model, derivation of the recovery quality performance ceiling, a two-group generalization of the EQO algorithm, and algorithm performance in other model ecological scenarios.
(PDF)

**S1 Fig. Example 2-group, 3-group and 4-group groupings under a 3-group ground truth with their scores shown on the right.**
(TIF)

**S2 Fig. Effect of "nonfunctional group" on function.** We consider the model in main text with $N = 2$. (A) The value of function (the final metabolite) shown against the abundance of the nonfunctional group (species not involved in producing this metabolite) for an example dataset of 900 samples. The scatter plot shows a negative correlation. The red line is the least-squares line and r marks the Pearson correlation. (B) The coefficients of all species of a $S$–dimensional regression of function against all $S$ species, for the same dataset as in (A). Error

bars indicate 95% confidence intervals for the coefficient estimates. The $x$ axis is ordered so that species 1–24 belong to the functional group (red) and species 25–48 belong to the nonfunctional group (blue). We see that most nonfunctional species have negative regression coefficients. (C) The grouping scores for the outputs of EQO, EQO-2g, K-Means and Metropolis algorithms over 50 simulated datasets (see S1 Text Section 3 for EQO-2g), shown as box plot with markers the same as Fig 2A and 2B. EQO-2g performs as well as Metropolis. Random groupings are included as controls. (D) Same as (C), with the competition strength tuned down by doubling the number of general resources. The performance of EQO is improved and comparable to other algorithms, as expected.
(TIF)

**S3 Fig. Metropolis is able to identify groups associated with functions other than end product of linear digradation chain.** The recovery quantity of each functional group as function of number of groups in output ($k$) for function to be (A) intermediate product of linear degradation chain; (B) one of the end product in a degradation chain with a branch; (C) common end product of 2 linear degradation chain. Groups are indicated in the pictogram of each panel. Numbers in the pictogram of (C) indicate the transfer ratio $w_r$ of each reaction. In the first two cases (A & B), Metropolis can identify all the functional groups. While in the last, it can only recover the group of direct producers.
(TIF)

**S4 Fig. Output groupings of the 3 algorithms for a linear degradation chain of $N = 3$ metabolites.** The output groupings of EQO, 2-group and 3-group K-means and Metropolis, as correspond to Fig 2A and 2B in the main text. Species 1–16 belong to group 1, 17–32 belong to group 2 (direct producers), 33–48 belong to group 3 (nonfunctional species).
(TIF)

**S5 Fig. Heat maps of mean scores over 50 datasets of the algorithms as a function of relative noise and number of samples.**
(TIF)

**S6 Fig. Recovery qualities of individual groups under increasing intra-group heterogeneity and inter-group promiscuity of species.** As an extension of the Fig 2D analysis in the main text, we look into the per-group recovery quality (defined in S1 Text Section 2) of the 3-group Metropolis, for the scenario of a linear degradation chain of $N = 3$ metabolites. (A) The recovery quality of the upstream group 1 and the direct producers group 2 shown as a function of intra-group heterogeneity (no promiscuity). (B) Same, as a function of inter-group promiscuity (no heterogeneity). Black dashed line is the random-group control (average quality of a 3-group random grouping).
(TIF)

## Acknowledgments

We thank A. Goyal, X. Shan, C. Holmes, J. Moran, F. Yu and Q. Wang for useful discussions.

## Author Contributions

**Conceptualization:** Otto X. Cordero, Mikhail Tikhonov.

**Data curation:** Yuanchen Zhao.

**Formal analysis:** Yuanchen Zhao.

**Funding acquisition:** Otto X. Cordero, Mikhail Tikhonov.

**Investigation:** Yuanchen Zhao.

**Methodology:** Yuanchen Zhao, Mikhail Tikhonov.

**Software:** Yuanchen Zhao.

**Supervision:** Mikhail Tikhonov.

**Writing – original draft:** Yuanchen Zhao.

**Writing – review & editing:** Yuanchen Zhao, Otto X. Cordero, Mikhail Tikhonov.

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
