## [Decision Letter · Decision Letter 0]

10 Jul 2024

Dear Mr Tikhonov,

Thank you very much for submitting your manuscript "Linear-regression-based algorithms can succeed at identifying microbial functional groups despite the nonlinearity of ecological function" for consideration at PLOS Computational Biology. As with all papers reviewed by the journal, your manuscript was reviewed by members of the editorial board and by several independent reviewers. The reviewers appreciated the attention to an important topic. Based on the reviews, we are likely to accept this manuscript for publication, providing that you modify the manuscript according to the review recommendations.

Dear Authors,

Thank you for submitting your manuscript to PLOS Computational Biology. After careful consideration of the feedback from the reviewers, we are pleased to inform you that your manuscript has strong potential for publication, pending minor revisions.

Overall, the reviewers found your work to be thorough, well-conducted, and biologically significant. They particularly appreciated the innovative approach you proposed for defining groups of organisms that perform specific subfunctions within a community metabolism, and the successful application of your model and algorithms to synthetic data. However, to enhance the clarity and impact of your manuscript, the reviewers have suggested several areas for improvement:

1. Introduction and Background: Expand the introduction to provide a more detailed summary of the ensemble quotient optimization (EQO) approach for readers who may not be familiar with your previous work. This will help to contextualize your current study better.

2. Figures and Presentation: Improve the clarity of Figure 1 to make the model assumptions more explicit. Consider using consistent coloring schemes to enhance understanding. Additionally, provide a clearer explanation of the growth function (eqn 1a) and its components, particularly the summation of resource utilization terms and the rationale behind not solely relying on the limiting resource.

3. Methodological Details: Include more detailed explanations of the efficiency of degradation in your equations and the reasoning behind your parameter choices to avoid overfitting. This will help readers understand the robustness of your model and the implications of your findings.

4. Abstract and Writing Style: Refine the abstract and other sections to avoid any language that may be perceived as puzzling or overly complex. Aim for straightforward and precise descriptions of your findings and their significance.

5. Data and Code Availability: Ensure that all data and computational code underlying your findings are made fully available in accordance with PLOS's data policy. This transparency is crucial for the reproducibility and credibility of your research.

We believe that addressing these points will significantly enhance the readability and impact of your manuscript. Please submit a revised version along with a detailed response to the reviewers' comments.

Thank you for your efforts, and we look forward to receiving your revised manuscript.

Best regards,

Leonardo Oña

Guest Editor, PLOS Computational Biology

Sincerely,

Leonardo Oña

Guest Editor

PLOS Computational Biology

Zhaolei Zhang

Section Editor

PLOS Computational Biology

Dear Authors,

Thank you for submitting your manuscript to PLOS Computational Biology. After careful consideration of the feedback from the reviewers, we are pleased to inform you that your manuscript has strong potential for publication, pending minor revisions.

Overall, the reviewers found your work to be thorough, well-conducted, and biologically significant. They particularly appreciated the innovative approach you proposed for defining groups of organisms that perform specific subfunctions within a community metabolism, and the successful application of your model and algorithms to synthetic data. However, to enhance the clarity and impact of your manuscript, the reviewers have suggested several areas for improvement:

1. Introduction and Background: Expand the introduction to provide a more detailed summary of the ensemble quotient optimization (EQO) approach for readers who may not be familiar with your previous work. This will help to contextualize your current study better.

2. Figures and Presentation: Improve the clarity of Figure 1 to make the model assumptions more explicit. Consider using consistent coloring schemes to enhance understanding. Additionally, provide a clearer explanation of the growth function (eqn 1a) and its components, particularly the summation of resource utilization terms and the rationale behind not solely relying on the limiting resource.

3. Methodological Details: Include more detailed explanations of the efficiency of degradation in your equations and the reasoning behind your parameter choices to avoid overfitting. This will help readers understand the robustness of your model and the implications of your findings.

4. Abstract and Writing Style: Refine the abstract and other sections to avoid any language that may be perceived as puzzling or overly complex. Aim for straightforward and precise descriptions of your findings and their significance.

5. Data and Code Availability: Ensure that all data and computational code underlying your findings are made fully available in accordance with PLOS's data policy. This transparency is crucial for the reproducibility and credibility of your research.

We believe that addressing these points will significantly enhance the readability and impact of your manuscript. Please submit a revised version along with a detailed response to the reviewers' comments.

Thank you for your efforts, and we look forward to receiving your revised manuscript.

Best regards,

Leonardo Oña

Guest Editor, PLOS Computational Biology

Reviewer's Responses to Questions

**Comments to the Authors:**

Reviewer #1: Review attached in separate file

Reviewer #2: The authors seek to define groups of organisms that perform specific subfunctions that are part of a community metabolism. They propose an approach where the organisms performing these subfunctions can be fit from data given a model. They propose a consumer-resource model where a community of microbes carries out a collective function. They create synthetic data with the model. They compare three algorithms to fit the groupings of the species based on the relative abundances and the function score. Biologically, this scenario is interesting because it provides a mechanism for community-level selection in microbial communities: if selection acts on the degradation (or synthesis) of a product, then all of the necessary species for its production are selected for as a group. Methodologically, this paper extends the EQO approach previously developed by the authors with a Metropolis algorithm-based fitting approach that correctly detects the groupings with good accuracy in the synthetic dataset. Overall, I am enthusiastic about the paper. The work appears thorough, and I have no major concerns. There were some aspects of the presentation that were a bit opaque because the authors refer to past work a lot without providing the specific pertinent details from the cited works. These instances can be improved with minor rewriting. I think the paper is a good fit for PLOS Computational Biology.

Here are some very minor suggestions to improve the manuscript. My guess is that some of these comments are based on my own misconceptions, and I hope the authors will improve the presentation so that readers do not come away with misconceptions.

. I appreciate the brevity but feel it would be helpful to expand the introduction to briefly summarize the ensemble quotient optimization approach in a bit more detail for the naiive reader. Some of this could potentially be accomplished in Figure 1 as well.

. Figure 1 is something of a graphical abstract intended to summarize the approach of the paper, but it does not make certain specifics clear enough. I wasn't sure from Fig. 1 what the model is assuming. If this is a linear pathway with intermediates, a product will not be synthesized unless all the intermediate steps are completed. Or is the idea that all enzymes in the pathway are present in each 15-member community, and it is the abundance of each of the 15 members that determines the flux through their step in the pathway? From the reading the whole paper, I assume the latter, but please clarify early on. Also, could the colors in 1a correspond to those in 1c? For instance, coloring by group in 1a could be one way to make it more consistent.

. I found the growth function (eqn 1a) a bit unintuitive the way it is presented, and I would like to see a clarification. Here is why I think I was confused: I am used to thinking of bacteria feeding on one limiting resource at a time, in the spirit of the lac operon in E. coli. From a consumer-resource perspective, Liebig's law of the minimum makes the non-limiting resources ignorable. In eqn 1a, the growth is a summation of two resource utilization terms balanced by the maintenance cost of the species. How should I think of this in terms of setting the growth rate? Are these two independent ways the population can grow? Why isn't just the limiting resource setting the growth rate? Should we think of a time-averaged growth rate where the growth on multiple limiting resources that fluctuate over time needs to be factored in?

. In equation 1a, first summand in the brackets; equation 1b, third summand; and equation 2, first summand: it appears the efficiency of degradation of a metabolite by a species is 100%? I would think there would be a rate constant, such that a fraction of the resource is consumed/converted. However, I also see that the realized growth is greater than 1 and there is a metabolic cost term, so it's not normalized to 1. Things probably balance out in the end. This seems like a good strategy to reduce the number of parameters to reduce overfitting. Was that the intent? Could the reasoning be explained a bit more in the text? I see references to previous papers, and these are nice papers, but explaining it a bit more here would be helpful.

. Very minor: End of page 4: Please add additional citations of the Materials and Methods section to the mention of each algorithm. It wasn't clear that the descriptions were there, and they greatly help.

. Certain embellishments are confusing. For example in the abstract, "This success seems puzzling, since modeling community function as a linear combination of contributions of individual species appears simplistic." I'm not sure why it seems puzzling when the function is governed by a linear pathway. Can the sentence be left out? The paper seems pretty straightforward, but the writing in certain cases alludes to something more mysterious, which is confusing.

Reviewer #3: This manuscript by Yuanchen Zhao, Otto Cordero and Mikhail Tikhonov applies several algorithms for functional group detection in ecological communities (including a newly proposed algorithm) to a simulated dataset of species abundances and measurements of a community-level functional trait. The authors show that several approaches, and especially their newly introduced method, can correctly recover the true functional grouping in the generated dataset, and that models based on straighforward linear associations can outperform more complex models with respect to the correct identification of functional groups. Based on these results, the authors provide important clues on the limitations of the different algorithms and their applicability in different contexts, which makes this manuscript interesting and relevant for a broader audience. However, I feel the authors could have gone a bit further in exploring the limits of applicability of functional group detection algorithms within the scope of their model. Another concern I brought up in my comments is that the assumption of near-neutrality of species within functional groups (species are interchangeable, except for their random consumption of generic resources) might have coloured the results, in particular with respect to the simultaneous detection of multiple functional groups from one-dimensional functional data. Because the manuscript for review did not have line numbers, I have included these and other remarks as comments in the attached annotated PDF.

**Have the authors made all data and (if applicable) computational code underlying the findings in their manuscript fully available?**

Reviewer #1: Yes

Reviewer #2: **No: **I didn't see data files or scripts in the submission, but I assume the authors can easily make those available.

Reviewer #3: Yes

PLOS authors have the option to publish the peer review history of their article (what does this mean?). If published, this will include your full peer review and any attached files.

Reviewer #1: No

Reviewer #2: No

Reviewer #3: No

Figure Files:

Data Requirements:

Reproducibility:

References:

---

## [Decision Letter · Decision Letter 1]

24 Oct 2024

Dear Mr Tikhonov,

We are pleased to inform you that your manuscript 'Linear-regression-based algorithms can succeed at identifying microbial functional groups despite the nonlinearity of ecological function' has been provisionally accepted for publication in PLOS Computational Biology.

Best regards,

Leonardo Oña

Guest Editor

PLOS Computational Biology

Zhaolei Zhang

Section Editor

PLOS Computational Biology

Feilim Mac Gabhann

Editor-in-Chief

PLOS Computational Biology

Jason Papin

Editor-in-Chief

PLOS Computational Biology

Dear Authors,

We are pleased to inform you that your manuscript has been accepted for publication in PLOS Computational Biology. The reviewers have appreciated the revisions you made, particularly the additional tests conducted to address their concerns. While one reviewer has expressed some remaining differences of opinion regarding the design of the RC model, they acknowledged that these points should not impede the publication process. The second reviewer commended your thorough response to their feedback and the improvements made to the manuscript. We look forward to seeing the future applications of your method, as mentioned in your manuscript.

Congratulations on this achievement, and we will proceed with the publication process.

Best regards,

Leonardo Oña

Guest Editor, PLOS Computational Biology

Reviewer's Responses to Questions

**Comments to the Authors:**

Reviewer #1: I appreciate that the authors further tested the limits of their model and my concerns have been mostly addressed. However, i am afraid there will be some disagreement left between the authors approach and my opinion. I still feel that fact that the RC model was designed in a way that the analysis kind of has to work not optimal and a little misleading. However, i do not want to further delay the publication of the manuscript.

Reviewer #3: The authors have given serious consideration to my review rapport, and included additional stress-tests to better delineate the conditions under which their method delivers a reliable functional grouping. I recommend to accept the manuscript for publication in its current form. Also, I look forward to see the application of this method to an empirical dataset that the authors promise for a future publication.

**Have the authors made all data and (if applicable) computational code underlying the findings in their manuscript fully available?**

Reviewer #1: Yes

Reviewer #3: Yes

PLOS authors have the option to publish the peer review history of their article (what does this mean?). If published, this will include your full peer review and any attached files.

Reviewer #1: No

Reviewer #3: No

---

## [Editor Report · Acceptance letter]

6 Nov 2024

PCOMPBIOL-D-24-00586R1 

Linear-regression-based algorithms can succeed at identifying microbial functional groups despite the nonlinearity of ecological function

Dear Dr Tikhonov,

I am pleased to inform you that your manuscript has been formally accepted for publication in PLOS Computational Biology. Your manuscript is now with our production department and you will be notified of the publication date in due course.

With kind regards,

Anita Estes
